# Compounded Effervescent Magnesium for Familial Hypomagnesemia: A Case Report

**DOI:** 10.3390/ph16060785

**Published:** 2023-05-24

**Authors:** Giada Bennati, Mario Cirino, Giulia Benericetti, Natalia Maximova, Monica Zanier, Federico Pigato, Anna Parzianello, Alessandra Maestro, Egidio Barbi, Davide Zanon

**Affiliations:** 1Department of Pharmacy and Clinical Pharmacology, Institute for Maternal and Child Health—IRCCS Burlo Garofolo, 34137 Trieste, Italy; 2Department of Pediatrics, Pediatrics, Bone Marrow Transplant Unit, Institute for Maternal and Child Health—IRCCS Burlo Garofolo, 34137 Trieste, Italy; 3Department of Medicine, Surgery and Health Sciences, Postgraduate School of Clinical Pharmacology and Toxicology, University of Trieste, 34127 Trieste, Italy; 4Department of Medicine, Surgery and Health Science, University of Trieste, 34127 Trieste, Italy; 5Department of Pediatrics, Institute for Maternal and Child Health—IRCCS Burlo Garofolo, 34137 Trieste, Italy

**Keywords:** galenic compound, pediatrics, magnesium supplementation, clinical pharmacist, FHHNC

## Abstract

Familial hypomagnesemia with hypercalciuria and nephrocalcinosis (FHHNC) is a rare autosomal recessive disorder affecting <1/1,000,000 people. It is caused by mutations in the CLDN16 (FHHNC Type 1) or CLDN19 (FHHNC Type 2) genes, which are located on Chromosomes 3q27 and 1p34.2, respectively. There are no drug therapies for this condition. Although magnesium salts represent an important class of compounds and exhibit various therapeutic actions as a supplement for magnesium deficiency in FHHNC, various formulations on the market have different bioavailability. We report the case of a patient with FHNNC first treated, in our Pediatric Institute, with high doses of magnesium pidolate and magnesium and potassium citrate. The patient began to neglect this therapy after experiencing frequent daily episodes of diarrhoea. Our pharmacy received a request for an alternative magnesium supplement that would better comply by ensuring a good magnesium intake which will result in adequate blood magnesium levels. In response, we developed a galenic compound in the form of effervescent magnesium. Here, we report on the promise of this formulation not only for better compliance than pidolate, but also for better bioavailability.

## 1. Introduction

Familial hypomagnesemia with hypercalciuria and nephrocalcinosis (FHHNC) is a rare genetic disease affecting <1/1,000,000 individuals; only about 200 cases have been identified worldwide. FHHNC is an autosomal recessive disease, classified as a monogenic disorder. It is caused by mutations in the CLDN16 (FHHNC Type 1) or CLDN19 (FHHNC Type 2) genes, located on Chromosomes 3q27 and 1p34.2, respectively. To date, up to 69 different mutations have been found in the CLDN16 gene, and 22 in the CLDN19 gene. Most of these are missense mutations found in the homozygous state. Because of the autosomal recessive pattern of inheritance, parental consanguinity is frequent [1] and, characteristically, a geographic distribution of genetic variants has been noted. The c.416C>T (p.A139V) mutation of the CLDN16 gene is highly recurrent in the north of Africa, while around two-thirds of Spanish/Hispanic patients carrying mutations on the CLDN19 gene exhibit mainly the c.59G>A (p.G20D) mutation in homozygosis [2]. The CLDN16 and CLDN19 genes encode for Claudin-16 and Claudin-19, respectively. The thick ascending branch of the loop of Henle is the primary site of magnesium reabsorption in the renal tubule. The primary abnormality in patients with the syndrome of familial hypomagnesaemia, hypercalciuria, and nephrocalcinosis is a defect in the reabsorption of magnesium and calcium at this level. The consequence is a prolonged and severe renal wasting of magnesium, accompanied by hypercalciuria. In contrast, the tubular manipulation of potassium and chloride is normal. Thus, hypokalemia is not observed in this entity [3]. The claudin proteins are tight junction proteins mostly sited in the thick ascending limb of Henle’s loop. Thus, they are involved in the renal reabsorption process (Figure 1). Claudin-16 and Claudin-19 are known to polymerize to form a complex that acts as a pore necessary for paracellular ion transport, mainly elemental magnesium (Mg^2+^). These proteins also regulate the calcium elemental (Ca^2+^) paracellular transport. Therefore, the claudin proteins are responsible for the Mg^2+^ and Ca^2+^ reabsorption process [4]. Because the pathogenic variants of CLDN16 and CLDN19 genes cause a partial-to-complete loss of claudin-16 and claudin-19 function, the renal reabsorption of Mg^2+^ and Ca^2+^ is reduced, followed by excessive urinary loss of these cations. As a direct consequence, mild-to-severe hypomagnesemia and hypercalciuria occur in patients with FHHNC [1]. Accordingly, medullary bilateral nephrocalcinosis is found in almost all patients at diagnosis. Nevertheless, it should be noted that there is great variability in the phenotype of the disease, even among patients carrying the same mutation. The most frequent clinical symptoms include polyuria and polydipsia, enuresis, urinary tract infections, vomiting, cramps, and abdominal pain. The diagnosis of FHHNC is generally based on such clinical symptoms, which often occur within the first 5 years of life, although they may be overlooked due to their non-specific natures [1]. As for the phenotype of patients carrying CLDN19 mutations, it is now well-established that it can be characterized by congenital ocular defects leading to variable visual impairment, such as myopia magna. This distinctive trait can be explained because Claudin-19 is also expressed in the retinal pigmental epithelium. For a more definitive diagnosis of FHHNC, additional laboratory findings are needed, i.e., hyperuricemia, elevated serum creatinine, and high PTH [5]. However, the only way to confirm the diagnosis with certainty is through genetic testing [6]. Once diagnosed, there is no specific therapy for FHHNC, but the main goal is supportive treatment, hydration, and prevention of the severe progression of kidney damage. Low doses of thiazides are used with caution in FHHNC patients to treat hypercalciuria and reduce the progression of nephrocalcinosis, while avoiding further kidney damage due to volume depletion. However, the long-term impact of this therapy is controversial. To date, Mg^2+^ oral supplementation seems to be the most suitable supportive treatment in patients with preserved renal function, while in the worst cases, kidney transplantation remains the only option. Typically, the daily dose in children is 10-to-20 mg of Mg^2+^ per kg of body weight (0.41–0.83 mmol Mg^2+^/kg), distributed in three doses. Doses are titrated based on related symptoms to avoid adverse events such as diarrhoea, abdominal pain, and other gastrointestinal effects [1].

## 2. Case and Methods

A 20-year-old woman, first diagnosed with FHHNC at the age of 14 months, was admitted to be evaluated for “possible improvements in her well-being.” The patient had received oral magnesium supplementation from the time of her diagnosis. Since then, she had suffered an episode of tetany, at 4 years of age, due to dyselectrolytemia. Episodes of paresthesia in the hands and feet occurred when she was 16 years old. Currently, she has had a daily intake of 21 sachets of magnesium and potassium citrate (90 mg of Mg^2+^ per sachet), plus 21 sachets of magnesium pidolate (184 mg of Mg^2+^ per sachet), amounting to almost 6 g of Mg^2+^ daily. As reported in the available medical reports, over the years, the nephrological condition has remained substantially unchanged both from a clinical and laboratory point of view. The orthopedic and ophthalmic conditions of the patient, typically affected by this disease, were also in the normal range [1]. In March 2021, at the follow-up at the Nephrology Service, the patient reported difficulty following the dosage regimen of magnesium supplementation due to the high number of daily doses and the high frequency of diarrheal episodes. As a result, she wanted to suspend the magnesium supplementation. Hospitalization of the patient was mandatory to control blood electrolyte levels and limit the progression of the disease while waiting for the Clinical Galenics service of our Hospital Pharmacy to provide an alternative to magnesium supplementation that was acceptable to the patient. The aim of this study was to compound a formulation with improved bioavailability and a reduction of side effects to increase compliance, rather than to effect a change in magnesium blood value.

A literature review on magnesium deficiency provided evidence that magnesium salts of organic acids and amino acids of magnesium, compared to salts of inorganic acids, have greater oral bioavailability and absorption. This is in large part because magnesium–amino acid compounds are transported into the cell through dipeptide channels, and there are many more dipeptide channels in the intestine than ion channels. Furthermore, inorganic magnesium salts that are not absorbed generally cause diarrhoea and laxative effects due to the osmotic activity in the intestine and colon and to the stimulation of gastric motility, worsening the patient’s quality of life [7,8]. Therefore, in April 2021, our hospital pharmacy developed a powder compound of dibasic magnesium citrate plus magnesium bisglycinate. The galenic compound produced by the hospital pharmacy was titrated, starting with two doses of 250 mg of Mg^2+^ on the first day. Then, on the second day, this was increased to 500 mg × 4 and finally increased to 4 g of Mg^2+^ per day. At a follow-up in June 2021, the physician certified good adherence to the new therapy with an improved quality of life. There have been no more episodes of lower extremity cramps requiring an additional dose of magnesium, nor the presence of gastrointestinal adverse effects. Blood tests confirm the maintenance of electrolyte parameters, specifically that magnesium persistence is low but stable and mainly increased after the switch to the galenic compound (Appendix A). We also developed a Visual Analog Scale (VAS) questionnaire (Appendix A) based on PedsQL 4.0, which was administered to the patient and to one of the parents, to evaluate health-related quality of life (HRQOL) in patients with chronic health conditions [9].

## 3. Discussion

The chemical form in which magnesium occurs is key to ensuring a known and constant bioavailability of the elemental ion. Organic and inorganic magnesium are the two main groups of magnesium compounds. The bioavailability of organic magnesium compounds is higher than that of inorganic forms. In particular, the bonds between magnesium and amino acids are stronger, more stable, and survive longer in the bloodstream. Galenic formulation also addresses the principles of drug preparation and coupling to optimize the absorption of compounds. Adequate galenic formulation is important in the absorption of magnesium compounds [10]. Currently, there are several oral preparations available on the market to supplement the intake of Mg^2+^ with the diet. They differ in the form of dosage and/or in the type of Mg^2+^ compound involved, which can be, as mentioned, inorganic or organic (Table 1).

Magnesium bisglycinate was chosen because it offers greater bioavailability, greater tolerability, and appears to be cleared from the plasma compartment more slowly. Data from in-vitro studies suggest that magnesium bisglycinate can be absorbed through dipeptide absorption pathways in the upper part of the small intestine [12]. Magnesium citrate was chosen because, in addition to its better absorption than inorganic salts [13,14], it is known to bind to oxalates in the intestinal tract, thereby decreasing its absorption and accumulation in the body. Since most kidney stones are calcium oxalate, reducing the oxalate content in the body has been found to decrease the risk of nephrolithiasis [15,16,17], a common occurrence in FHHNC. We report a clinical case of FHHNC in which the patient, after years of magnesium supplementation, showed a worsening of adherence to treatment due to the high amount of magnesium in the daily doses and the onset of diarrhoea, a side effect that usually occurs usually with high doses of magnesium [18]. Despite the current knowledge of the pathogenesis of the disease, specific treatment is lacking. Supportive treatment, hydration, and prevention of acquired kidney damage remain the only option to delay the progression for those patients with preserved kidney function. Based on local availability, different magnesium salts are prescribed for the treatment of individuals with significant hypomagnesemia. Because most patients cannot be expected to achieve the complete correction of serum magnesium levels, doses are generally titrated according to the presence and severity of related symptoms to avoid adverse events at high doses of magnesium such as diarrhoea, abdominal pain, and other gastrointestinal side effects [1]. The use of magnesium citrate salts, which bind urinary oxalates, may be useful in slowing the development of nephrocalcinosis. In vivo, oxalates have two negative charges and magnesium ions have two positive charges, so oxalates easily deplete magnesium from the body; the presence of Mg^2+^ reduces the average size of the calcium oxalate. This effect is determined to be Mg^2+^ concentration-dependent. It seems that Mg^2+^ inhibition is synergistic with citrate. The presence of magnesium ions tends to destabilize calcium–oxalate ion pairs and reduce the size of their aggregates [19]. However, in the references, it is not clear how much magnesium binds the oxalate and how much will be reabsorbed into circulation and acting as a magnesium supplement [15]. The Clinical Galenic Service of the Hospital Pharmacy of our Institute has solved the problem by formulating a compound of magnesium bisglycinate in powder form with suitably flavored magnesium citrate, which has made it possible to reduce the number of daily doses while maintaining all stable clinical parameters, reducing side effects, and improving compliance and quality of life. These results are supported by the results of the VAS questionnaire, which is a useful tool for documenting symptoms and assessing disease control due to its simplicity, speed of management, and low susceptibility to errors [20,21]. Blood tests confirm the maintenance of the electrolyte parameters. In particular, the persistence of magnesium is low but stable and mainly increased after switching to the galenic compound. From the data available in the medical record, from the moment the patient was taken in charge by our hospital, no important changes in renal function were highlighted; parathyroid hormone (PTH) has always remained elevated. As reported in the available clinical documentation, there is no evidence of progression of nephrocalcinosis. Physicians describe “marked hyperechogenicity of the pyramids due to nephrocalcinosis,” which remains constant throughout the patient’s observation period; even bone mineral density, despite remaining low, does not undergo significant alterations. The daily excretion of magnesium in the urine also remains constant, showing peaks in correspondence with intakes of higher doses of magnesium [Appendix A. When analyzing the VAS questionnaire [Appendix A, we can see how the quality of life, reported by both the patient and the parent, has considerably improved after the therapeutic switch to the galenic compound. In fact, while the previous therapy with commercial products had a negative effect on adherence, compliance, and judgment of general health, these factors considerably improved following the start of therapy with the galenic preparation. Furthermore, as reported in the patient’s clinical documentation, the neurological problems (paresthesia, headache, sleep disorders, etc.) also resolved after the start of treatment with the galenic compound of Mg^2+^.

## 4. Conclusions

To date, there are no published guidelines for the management of patients with FHHNC. To our knowledge, oral magnesium supplementation seems to be the most suitable supportive treatment in patients with preserved renal function. Magnesium compounds based on organic salts and amino acids (citrate and bisglycinate in particular) are characterized by high bioavailability and low side effects, and these characteristics have been confirmed in this case. The Hospital Pharmacy can develop a galenic compound that satisfies the patient’s needs with respect to high therapeutic adherence, a good quality of life, and guaranteed high therapeutic efficacy.

## Figures and Tables

**Figure 1 pharmaceuticals-16-00785-f001:**
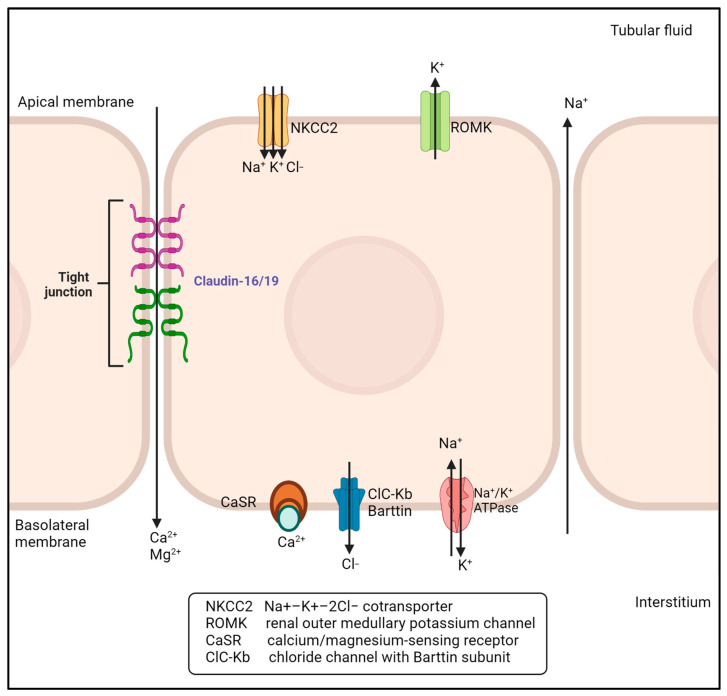
Paracellular reabsorption of magnesium and calcium in the thick ascending limb of Henle’s loop. Claudin-16 and Claudin-19 facilitate the paracellular transport of magnesium and calcium.

**Table 1 pharmaceuticals-16-00785-t001:** Bioavailability/Pharmacokinetics of Magnesium (V.V. Ranade et al. [11] American Journal of Therapeutics 2001).

Magnesium Salts	Carbonate	Chloride	Citrate		Fumarate	Gluconate		Glycinate	L-lactate
**Elemental Mg^2+^/dose,** **mg (mEq)**	232 (19.0)	64 (5.26)	- (25)		530 (44.16)	Tablet 27 (2.2)	Liquid 54 (4.4)	100 (8.33)	84 (7)
**Solubility in water**	Nearly insoluble	High	Very good		Good	Moderate		Good	Excellent
**Bioavailability**	Extremely low	Good	Good		Good	Good; similar to chloride		Good	Excellent
**Oral absorption,** **% (mEq)**	-	19.68 (1.04)	29.64 (ionic)		-	19.25 (0.82–0.43)		23.5 -	42.3 (2.96)
**Delivery system**	Tablets	Enteric coated Tablets	Liquid, Tablets		Tablets	Tablets, liquid		Ingestion	Sustained-release caplets
**Dosage**	70 mg elemental Mg (each Tablet)	640 mg/d, 1–2 tabs TID	25 mEq Mg, 2–5 Tablets		1 Tablet	648 mg/d, 2–4 Tablets TID		100 mg	1–2 caplets q 12 h
**Side effects**	GI distress, diarrhea	GI distress, diarrhea	Laxative, evacuant			GI distress, diarrhea			Minor GI disturbances
**Comments**	Not very soluble at pH of GI tract; some GI side effects; laxative	Enteric coating could delay absorption; some GI side effects; cathartic	Therapy-limiting side effects; limited absorption; low citraturic response			Expensive formulation to achieve Recommended daily allowance requirements		Good alternative in patients with intestinal resection	Sustained release increases absorption, reduces side effects; cathartic
**Magnesium salts**	**Oxide**	**K Mg** **citrate**	**DL-aspartate**	**L-aspartate**	**Hydroxide**	**Salicylate**		**Sulfate**	**Aminoate**
**Elemental Mg^2+^/dose,** **mg (mEq)**	241 (19–8)	— (24.5)	5	5	2 × 10.3 mmol	600		56.5 mmol	500 (41.6)
**Solubility in water**	Extremely low, 8.6 mg/L	High solubility	Good	Good	Practically insoluble	Freely soluble		Moderately soluble	
**Bioavailability**	Extremely Low	Good; similar to Mg citrate				86–100%			
**Oral absorption,** **% (mEq)**	22.8 (0.39) (2% ionic)		44.5	41.7				4 (oral dose), limited and variable extent	
**Delivery system**	Tablets, capsules	Tablet	Tablet	Tablet	Tablet (Maalox)	Tablet		IV solution	Tablet
**Dosage**	2–4 tabs TID	7 tablets, 3–5 mEq Mg ea	1 Tablet	1 Tablet	2 Tablets	600 mg, 1 Tablet		Intravenous Mg 9.9–49.3 mg/mL	1 Tablet, 3 Tablets (100 mg ea Mg)
**Side effects**	Emesis, diarrhea	No GI side effects			Occasional regurgitation and mild diarrhea				
**Comments**	Virtually insoluble at pH of GI tract; some GI side effects; antacid	Yielded a greater Citraturic response in addition to primary absorbable K & Mg			Antacid; cathartic	Internal, antiinfective		Parenteral use may lead to magnesium toxicity	

Abbreviations: GI, gastrointestinal; K, potassium mEq, milliequivalent; Mg, magnesium; TID, ter in die.

## Data Availability

No new data were created.

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
