# Peer review of "Compounded Effervescent Magnesium for Familial Hypomagnesemia: A Case Report"

_pharmaceuticals, 2023, doi:10.3390/ph16060785_

Round 1
Reviewer 1 Report
The authors described a patient with familial hypomagnesemia with hypercalciuria and nephrocalcinosis (FHHNC) and was treated with magnesium supplementary therapy. Characteristics of FHHNC was reviewed and regimens of magnesium supplement were reviewed. The authors concluded that the home-made powder compound: dibasic magnesium citrate plus magnesium bisglycinate was a good formula for magnesium supplement in their patient. There are several concerns for their report:
1. A more realistic calculation such as daily urine magnesium excretion is needed for their patient for better understanding the homeostasis of magnesium.
2. One of the major concerns is the long-term effects. The changes in renal function, electrolytes, parathyroid hormone, vitamin D and progression/regression of nephrocalcinosis, bone mineral density.
3. Totally 15 available regimens were listed and compared in table 1. However, no references were provided. In order to review this data, scientific data is mandatory.
4. The authors mentioned that magnesium citrate may bind to oxalate and then reduce the possibility of nephrocalcinosis. In this way, how many magnesium goes to bind oxalate and how many will be reabsorbed into circulation and acting as magnesium supplement? A more realistic description is required.
No
Reviewer 2 Report
Very well prepared article. The information is very useful and helpful for the clinical practice. On the one hand it provides concise but very comprehensive information about Familial hypomagnesemia with hypercalciuria and nephrocalcinosis (FHHNC) that is a very rare genetic disorder and each case report is of a contributory nature. On the other hand, magnesium replacement therapy options are reviewed and a very detailed and precise comparison table is presented. Magnesium compound based on organic salts and amino acids (citrate and bisglycinate) is used in this case that is with high bioavailability and low side effects.
In summary, I conclude that this case report will be at great interest and value to the readers.
One technical correction: “...“
admitted to be evaluated for “possible improvements in her well-being. The patient had - 82
